# Modified-Active Release Therapy in Patients with Scapulocostal Syndrome and Masticatory Myofascial Pain: A Stratified-Randomized Controlled Trial

**DOI:** 10.3390/ijerph18168533

**Published:** 2021-08-12

**Authors:** Wilawan Kanhachon, Yodchai Boonprakob

**Affiliations:** 1School of Physical Therapy, Faculty of Associated Medical Sciences, Khon Kaen University, Khon Kaen 40002, Thailand; wilawan.k@kkumail.com; 2Research Institute for Human High Performance and Health Promotion, Faculty of Medicine, Khon Kaen University, Khon Kaen 40002, Thailand

**Keywords:** myofascial linkage, upper quarter pain, scapular pain, neck pain, myogenic-temporomandibular disorder

## Abstract

Modified-active release therapy (mART) was developed to treat patients experiencing upper quarter pain. The objective of the study was to determine the effectiveness of the mART in treating pain, promoting function, and measuring emotions in patients with scapulocostal syndrome (SCS) and masticatory myofascial pain (MMP). A stratified-randomized controlled trial was employed in 38 participants separated into two groups. All participants underwent the same series visual analog scale (VAS), pressure pain threshold (PPT), mouth opening (MO), maximum mouth opening (MMO), craniovertebral angle (CV-angle), and pain catastrophizing scale Thai version (PCS-Thai-version) at the baseline. The mART group underwent the mART program three times a week for 4 weeks with a hot pack and an educational briefing while the control group received only a hot pack and the educational briefing. After treatment, both groups showed significant improvement (*p* < 0.05) in all parameters except MO, MMO, and CV-angle. When comparing outcomes between the groups, the mART group showed a statistically significant greater number of improvements than did the control group. In conclusion, the mART program can improve pain experienced by patients with SCS and MMP and it can be used as an adjuvant technique with conservative treatment.

## 1. Introduction

The human body’s largest organ is the skeletal muscle set, since it accounts for approximately 50% of body weight [1]. Likewise, the World Health Organization (2003) states that musculoskeletal pain is a major cause of morbidity [2] which is a growing financial burden on the healthcare system [1]. Musculoskeletal pain presents itself as local and referred pain [3]. Local pain is located closed to the source, whereas, referred pain is located in a different region away from the source—mostly associated with a central mechanism [4,5] and usually occurring as a chronic condition [6].

In what is referred to as the upper quarter of pain syndrome, neck pain, scapular pain, and temporomandibular joint (TMJ) pain are the three most common areas associated with chronic symptoms [7]. Various studies have looked at the relationship between neck pain and scapular pain [8,9,10,11] and between neck pain and MMP or myogenic-temporomandibular disorder (myogenic-TMD) [12,13,14,15,16,17,18]. These were conducted primarily to elucidate the effectiveness of treatment programs among these two association areas [19,20,21,22]. Knowledge concerning myofascial linkage also revealed the possible association between SCS and MMP through the attachment of the splenius capitis and rhomboid muscles which is part of the spiral line (SPL) along with the nuchal line [23]. In 2021, Kanhachon and Boonprakob investigated the correlation between medial scapular muscle pain and jaw function in patients with Scapulocostal syndrome (SCS). The results demonstrated that medial scapular muscle pain tends to influence maximum mouth opening. The authors recommended that therapists should assess the distance of maximum mouth opening amid treatment of patients with SCS [24]. Therefore, the present study is interested in investigating the effectiveness of treatment programs which were originally designed to treat patients with SCS and MMP.

Modified-active release therapy (mART) is a sub-type of movement therapy which combines of passive and active exercises. It comprises of a group of eight exercise positions for the jaw, neck, and scapular areas. The aim of mART is to have the active movements performed by the patients and passive movements done by the therapist cooperatively. mART was developed from the combination of Anatomy training [23], biopsychosocial context [25], mindfulness exercises [26], and the spinal stabilization system [27]. mART was originally designed to enhance the neuroplasticity through the new and complex dual tasks, disturb pain by signaling and enhancing the function of inhibitory descending pathways through manual techniques, reduce anxiety via mindful exercise, decrease fear of movement, and encourage self-efficacy during movement through a pain-free range of exercises. The mART program in this study was aimed at treating patients with SCS and MMP with a focus on pain, function, and emotion impairment. Furthermore, the program may be modified to manage other areas of the human body.

## 2. Materials and Methods

### 2.1. Study Design and Participants

This study incorporated a parallel design with stratified-randomized (block 4 and 6) control trial on thirty-eight academic participants (4 males and 34 females with ages ranging from 25 ± 5 years). There were a greater number of female participants than males due to the fact that the evidence reported that both SCS and MMP were most commonly found among females as opposed to males [28,29]. This stratified-randomized pain intensity (distance of VAS line) observation of the most painful area in the medial scapular muscles and masticatory muscles was carried out to determine pain intensity (45 to 74 mm and 75 to 100 mm). A single-blinded (assessor-blinded) study was conducted from 1 March to 15 December 2020 at the AMS wellness-health center at the faculty of Associated Medical Sciences, Khon Kaen University, Khon Kaen, Thailand. The inclusion criteria consisted of experienced pain at the neck, scapular, and jaw regions during sustained posture of more than 3 months, reported pain with a referral pattern while receiving manual pressure of at least 3 points on the medial scapular muscles (levator scapulae, rhomboid major, rhomboid minor, and serratus posterior superior), reported pain with referral patterns while receiving manual pressure applied to at least 1 point of the masticatory muscle (masseter, pterygoids, and temporalis), and presented pain with limited mouth opening distance (<40 mm) or presented with normal mouth opening distance (40–60 mm) with pain in the masticatory muscle. The exclusion criteria consisted of a reported a history of TMD with disc displacement and osteoarthritis, reported history of serious systemic diseases, reported history of serious conditions such as a spinal fracture or any inflammatory diseases, reported history of migraine with aura, tension-type headaches and cervicogenic headaches. Participants who passed the inclusion and exclusion criteria were asked to draw a lot from the concealed envelopes and were then assigned to the treatment group (mART) or the control group by the researcher. The education of home-physical therapy has been applied as the gold standard of treatment in participants with chronic pian [30]. Therefore, all participants in this study were taught on the basic anatomy of the scapula, TMJ, biomechanics, muscle attachment, dysfunctions which could occur in these areas, and the role of emotional factors in SCS and MMP.

Participants in the control group had a hot pack applied to their jaw and scapular areas for 15 min in addition to receiving the education on SCS and MMP for 15 min, respectively. They were asked to participate in the protocol 3 times a week for 4 weeks. Moreover, they did not receive any mART program related treatment. Details of the subject recruitment and allocation process of this study are summarized in Figure 1. This study was approved by the Khon Kaen University Ethics Committee for Human Research (HE612318) and registered at the Thai Clinical Trials Registry (TCTR20190512001). All participants provided informed consent prior to participating in the study.

### 2.2. Modified-Active Release Therapy Program

The mART program contained 8 exercise positions which were designed to encourage the motion of the jaw, neck, and scapular areas. Each exercise position combined the active movement by participants in conjunction with the passive movement performed by the therapist. During the exercise, participants were asked to hold each position for 6 s with 10 repetitions at each position, 8 positions per set, and 3 sets per day. These exercise movement patterns are described in Appendix A.

### 2.3. Interventions

All participants in the mART group participated in the exercise program for 4 weeks. The program was conducted three times per week for 60 min per session. The program consisted of a hot pack on the jaw and scapular areas for 15 min, an educational briefing of SCS and MMP for 15 min, and the mART program for 30 min. Participants were taught to perform each exercise correctly by the therapist prior to the start of the exercise program.

### 2.4. Outcome Measurement

The pain and functions’ outcomes of the participants were assessed at the beginning, immediately after the start of the program, at 4 weeks, and finally with a 1-month follow-up post program completion. Emotional outcomes were assessed at the beginning, at 4 weeks, and also at 1-month follow-up to the exercise program. Outcome assessors were blind to information about which group each participant belonged to. Participants received an explanation of the test and performed the test over two trials with 2–3 min rest between each test in order to reduce copying.

#### 2.4.1. Pain Intensity

The Visual Analog Scale (VAS) was used to assess pain intensity of the most painful areas of the medial scapular muscles and the masticatory muscles. Those painful areas were marked on clear-plastic sheets by the researcher to guide the testing area for the assessor. For the assessment of the scapular area, the assessment position required participants to sit with his/her hand on the opposite shoulder to the affected side. The assessor placed his hand on the medial scapular region to investigate the most painful area. In the jaw area, the assessment position required participants to sit in a relaxed position while the assessor placed his hand on the jaw of the affected side. VAS was demonstrated with high intra-rater reliability (ICC = 0.923 for SCS and ICC = 0.981 for MMP, respectively). The assessor applied a compression force of 2 kg/cm^2^ at the medial scapular muscles and 1 kg/cm^2^ at the masticatory muscles. Participants were asked to mark a short vertical line on the 100 mm horizontal VAS [31] to indicate pain intensity for both areas. The test was performed three times and presented as mean value for statistical analysis.

#### 2.4.2. Pressure Pain Threshold

The pressure algometry was utilized to measure the pressure pain threshold of all participants. The locations of pain, participant position, and position of assessor were the same as in the assessment of VAS. PPT was demonstrated with high intra-rater reliability (ICC = 0.969 for SCS and ICC = 0.987 for MMP, respectively). Moreover, it is also associated with catastrophizing, depression, and fear of movement [32]. While receiving the compression force, participants told the assessor to ‘stop’ at the amount of force which made them begin to feel pain or discomfort to indicate the pressure pain threshold for both areas. The tests were performed three times and presented as mean value for statistical analysis.

#### 2.4.3. Mouth Opening and Maximum Mouth Opening Distances

The TheraBite^®^ was employed to measure the distance of mouth opening [33,34]. For the MO condition, participants were asked to open their mouths without experiencing pain or discomfort in their masticatory muscles, while in the MMO condition. Participants were requested to open their mouths as wide as possible even if they felt pain in the masticatory muscles. The assessor measured the distance between the incisal edges of the upper and lower central incisors and recorded in millimeters. Mouth opening distance was demonstrated with high intra-rater reliability for both conditions (ICC = 0.965 for MO, and ICC = 0.846 for MMO, respectively). The test was repeated three times with 2 min rests between each test.

#### 2.4.4. Craniovertebral Angle

A three-point marker detection software method was employed to measure the CV-angle of all participants, which measured from the relative C7 spinous process, tragus of the ear, and acromion process [35]. CV-angle was demonstrated with excellent validity (r = 0.90, *p* = 0.01) when compared with the goniometer [35], and high intra-testing reliability (ICC = 0.907). The assessment position required participants to stand with head-on-trunk and shoulder alignment while looking straight ahead. The digital camera was located 1 m away from participants. Images were taken after participants performed the procedure five times on the spot [36]. The test was repeated three times with the mean value of degree of angle recorded.

#### 2.4.5. Pain Catastrophizing Scale

The PCS is defined as an exaggerated negative mental set brought to bear during the actual or anticipated painful experience of pain [37]. PCS is related to pain intensity, cognitive, disability, and quality of life among chronic pain patients [38,39,40]. This questionnaire contained 13 questions with a total possible score of 52 points. Interpretation was considered via the score; high scores were correlated with greater levels of pain catastrophizing [41]. A Thai version of the PCS questionnaire was translated and evaluated for validity and reliability of the test for use in the OA knee patients [42] which renders excellent reliability (ICC = 0.757 to 0.914). Likewise, this study presented excellent test–retest reliability (ICC = 0.990). Participants were asked to complete the questionnaire prior to receiving the intervention, following the last intervention, and at a 1-month follow-up. The test was performed twice with the mean value of the total score was recorded.

#### 2.4.6. Statistical Analysis

Sample size and percentage drop-out rate were calculated from a previous study which compared the effectiveness of home-PT (physical therapy) by itself with home-PT in conjunction with manual therapy for 4 weeks in participants with TMD [43]. The clinical acceptable margin of pain intensity was set at 2 points [44], with statistical power as 80%, and drop-out rate as 20%. This study incorporated 19 participants in each group.

Descriptive statistics were used to describe baseline demographics, clinical characteristics, and study findings. The Kolmogorov-Smirnov calculation was employed to ensure the normal distribution of the data. Two-way mixed ANOVA was used to determine the differences in all parameters at pre-test, immediately, at 4-weeks, and 1-month follow-up stages in each group.

Differences were considered at *p* < 0.05 level. All calculations were performed with SPSS version 23 (SPSS Inc., Chicago, IL, USA).

## 3. Results

Participant’s demographic data and health status were presented as mean and standard deviation as shown in Table 1. There were thirty-eight participants in the mART and control groups, with the average ages (mean ± SD) of 26 ± 5 years and 25 ± 5 years, respectively. In the mART group, the number of females and males totaled 19 participants (two males) with average weight, height, and body mass index (58.74 ± 11.69 kg., 164.16 ± 7.69 cm, 21.72 ± 3.39 kg/m^2^, respectively). In the control group, the number of females and males accounted for 19 participants (of which two were male) with average weight, height, and body mass index (54.32 ± 10.58 kg., 160.47 ± 6.62 cm, 21.04 ± 3.50 kg/m^2^, respectively). Participant’s demographic data did not differ between the two groups. Most participants were single women, mean working duration was 4 to 5 years, and mean of chronic pain duration was 2 years. In terms of other variables, most factors were found to be equally balanced between the two groups.

### 3.1. Pain Intensity

A within-group comparison of VAS of SCS, mART group revealed a significant decrease in pain intensity at the immediate stage (4.33 ± 1.68; *p* < 0.01), and 1-month follow-up (2.25 ± 1.74; *p* < 0.01) when compared with baseline data (4.98 ± 1.69). A control group also revealed a significant decrease in pain intensity at the immediate stage (5.84 ± 2.18; *p* < 0.01) and 1-month follow-up (3.75 ± 2.34; *p* < 0.01) when compared with baseline data (5.71 ± 1.85). Regarding in between-group comparison, the mART group demonstrated an improvement greater than the control group (*p* = 0.021).

According to within-group comparison on VAS of MMP, the mART group revealed a significant decrease at pain intensity at 4 weeks (2.02 ± 1.47; *p* < 0.01), and 1-month follow-up (2.90 ± 1.86; *p* < 0.01) when compared with baseline data (4.71 ± 1.70). The control group also revealed a significant decrease in time scores testing at 4 weeks (3.88 ± 2.76; *p* < 0.01), 1-month follow-up (3.23 ± 2.24; *p* < 0.01) when compared with baseline data (5.78 ± 2.27). With regards to in between-group comparison, the mART group exhibited an improvement greater than the control group (*p* = 0.047) (Table 2).

### 3.2. Pressure Pain Threshold

Concerning within-group comparison on PPT of SCS, the mART group revealed a significant increase in the pressure pain threshold at 4 weeks (2.28 ± 0.86; *p* < 0.05) when compared with baseline data (1.56 ± 0.90). The control group also revealed a significant increase in pressure pain threshold PPT at 4 weeks (1.72 ± 0.50; *p* < 0.05) when compared with baseline data (1.45 ± 0.58). Regarding between-group comparison, the mART group demonstrated an improvement greater than the control group (*p* = 0.030).

In terms of within-group comparison on PPT of MMP, the mART group showed a significant increase in the pressure pain threshold at 4 weeks (1.03 ± 0.26; *p* < 0.05) compared with baseline data (0.56 ± 0.37). The control group also revealed a significant increase in pressure pain threshold at 4 weeks (0.77 ± 0.26; *p* < 0.05) when compared with baseline data (0.57 ± 0.31). For in between-group comparison, the mART group displayed an improvement superior to the control group (*p* = 0.064) (Table 2).

### 3.3. Mouth Opening and Maximum Mouth Opening Distance

A within-group comparison of MO and MMO revealed no significant difference in mouth opening distance when compared with baseline for both groups. Likewise, in a between-group comparison, the mART group did not show statistically significant differences in both conditions (*p* = 0.573, and *p* = 0.707 for MO and MMO, respectively) (Table 2).

### 3.4. Craniovertebral Angle

A within-group comparison of CV-angle amid both groups revealed no significant difference in craniovertebral angle when compared with the baseline. Likewise, a between-group comparison of the mART group did not show statistically significant differences in either for CV-angle (*p* = 0.263) (Table 2).

### 3.5. Pain Catastrophizing Scale

With regards to a within-group comparison on PCS, the mART group revealed a significant decrease in pain on the catastrophizing scale at 4 weeks (12.35 ± 10.11; *p* < 0.001) and 1-month follow-up (10.46 ± 10.45; *p* < 0.001) when compared with baseline data (22.16 ± 9.33). A control group also revealed a significant decrease in pain catastrophizing at 4 weeks (13.16 ± 9.77; *p* < 0.01) and the 1-month follow-up (12.42 ± 10.49; *p* < 0.01) on comparison with baseline data (20.24 ± 11.41). In between-group comparison, the results did not show a statistically significant difference (*p* = 0.925) (Table 2).

## 4. Discussion

The objective of the study was to determine the effect of the mART program on pain, function, and emotion impairment in patients with SCS and MMP. The results revealed a significant improvement in pain intensity, pressure pain threshold, and pain catastrophizing subsequent to receiving the program in both groups. Additionally, the mART program seemed to show an improvement greater than those in the control group. However, MO, MMO, and CV-angle did not demonstrate any clinically significant changes among the two groups.

### 4.1. Pain Improvement in Patients with SCS and MMP

Pain intensity and pressure pain threshold were significantly improved in both groups. In addition, mART pointedly improved in both parameters greater than the control group. The possible mechanism of mART on pain improvement was explained by the signaling pathways of myofascial trigger points and therapies as described by Jafri in 2014 [45]. This hypothesis mentioned the initiation of mTrPs by a combination of chronic load on the muscle and psychological stress. The chronic load on the muscle caused microtubule proliferation which increased ROS (reactive oxygen species) and decreased ability of removal. Meanwhile, psychological stress contributes by reducing mitochondria content and increasing ROS production in cells. Prolonged muscle contraction restricted blood flow and initiated local ischemia resulting in muscle damage and an inflammatory response. Therefore, the recommendation of treatment on the mTrPs based on this hypothesis is to increase blood circulation and reduce ROS. Moreover, the systematic review suggested that a combination of stretching and strengthening exercises seemed to achieve greater effect than other types [46] because stretching and strengthening exercise improved blood flow and energy metabolism in muscles and reorganized muscle fiber cytoarchitecture [45,47]. Moreover, this type of exercise can be combined with other interventions, such as laser and manual therapy, to achieve greater outcomes in myofascial pain patients [48].

When considered, the identity of mART concerns manual therapy in conjunction with and encouraging active movement by patients in a pain-free range based on the function of myofascial linkage. In the other words, mART is a combination of stretching and strengthening exercise. Hence, mART programs may achieve the desired effect on pain reduction by the direct management of mTrPs, which increase blood circulation and reduce ROS. Additionally, there is some evidence mentioned that pain caused alteration of movement patterns and motor control [49], while the other evidence mentioned that changing of movement patterns caused pain [50]. However, Sahrmann (2010) summarized both concepts and emphasized that the treatment required reduction of pain, improvement in movement pattern, and motor control. Therefore, in the case of pain caused by alteration of movement patterns, mART can be an alternative method for the prevention of further problems. On the other hand, in the case of movement patterns that cause pain, the correction of movement patterns can alleviate symptoms [50].

### 4.2. Functional Improvement in Patients with SCS and MMP

A within-group comparison revealed that mouth opening distance and CV-angle did not show any significant differences among both groups. This result may be explained by the knowledge of neuromuscular adaptation of exercise which demonstrated that neural adaptation occurred before the muscular adaptation [51] i.e., the decrease in pain may not affect the muscular performance. What is more, previous studies suggested that the magnitude of muscle adaptation depended on type, intensity, frequency, and duration of exercise [52]. Since the mART program incorporates pain-free exercise and is performed for only four weeks, it may not reach the point of muscular adaptation. That is to say, the results did not reveal functional improvement, especially on the CV-angle which is associated with an imbalance between neck flexors and extensors over a long period.

Moreover, certain studies mentioned that CV-angle was correlated with myopia and neck posture. Some studies stated that the presence of forward head posture (FHP) in young adult participants accounted for 58.95%. Nonetheless, FHP was commonly found in females when compared with males (73.21% for females, and 26.79% for males, respectively). This showed individuals who experienced myopia ranged from 0 to −1.5D, with the greatest prevalence of FHP at (67.85%) [53]. Some studies investigated neck posture related to FHP by implementing two types of monitor: fixed and moving monitor. CV-angle analysis demonstrated that frequency of FHP was lower in the moving monitor than that in the fixed monitor. Hence, it could be concluded that sustained gazing at a fixed monitor can induce FHP [54]. In the present study, the author did not enquire about myopia among participants; and individuals who wore glasses were asked to remove them prior to CV-angle measuring. Therefore, vision impairment might have affected degree of CV-angle. Moreover, most participants were students and office workers who continuously worked on a fixed monitor such as a desktop computer or laptop during study participation. As a result, CV-angle was unchanged among participants.

Besides that, a study mentioned that a CV-angle less than 43.7 degrees was associated with ROM of MMO (ICC = 0.94) limit. It has been stated that abnormal postural origin may cause cumulative muscular and ligamental microtrama in the TMJ [55]. In the present study, the author revealed that average CV-angle post treatment was 38.11 degrees—which may affect the distance of mouth opening. In addition, in the case of distance of mouth opening, it was recorded that it had its own ceiling effect (40 to 60 mm for normal range). Since the author found the pain average of MMP was a score of about 4, which represents moderate pain, the moderate severity of pain among these participants may not affect the limitation on mouth opening distance, resulting in unchangeability of MO and MMO after receiving the mART program. What is more, mouth distance was represented as both an objective and subjective method since participants were asked to stop opening their mouth when they felt pain; consequently, it may have been influenced by other factors from the daily emotions of participants associated with the pain perception of pain.

### 4.3. Emotional Improvement in Patients with SCS and MMP

The pain catastrophizing scale was significantly improved in both groups which is consistent with pain improvement as described above. Likewise, the previous study mentioned that the whole PCS score correlated with pain intensity (r = 0.26) [56]. The degree of severity of PCS was interpreted via the score of questionnaire scores with higher scores representing greater severity. PCS score is defined as individuals’ exaggeration of pain perceptions associated with lack of emotional regulation ability to deal with pain [41]. As a result, individuals with chronic pain exaggerated negative responses toward actual pain. Even so, the average PCS score among participants was less than 24 points, which represented a mild level of PCS demonstrating that mART programs can yield better results than the baseline. According to this improvement, the author recommends that the mART program can promote a good functional health status and increase the ability to perform normal daily activities in patients with SCS and MMP.

## 5. Conclusions

The purpose of the study was to determine the effectiveness of the mART program in participants with SCS and MMP aged between 18 and 50 years old. After undergoing the program for 4 weeks (3 times/60 min/week), participants reported a greater positive impact with regards to pain (pain intensity and pressure pain threshold) and pain catastrophizing than those who only received hot packs and an educational briefing. Additionally, the mART program aimed to treat patients based on anatomy train concepts; therefore, the development of treatment programs for other symptoms is of potential interest.

## Figures and Tables

**Figure 1 ijerph-18-08533-f001:**
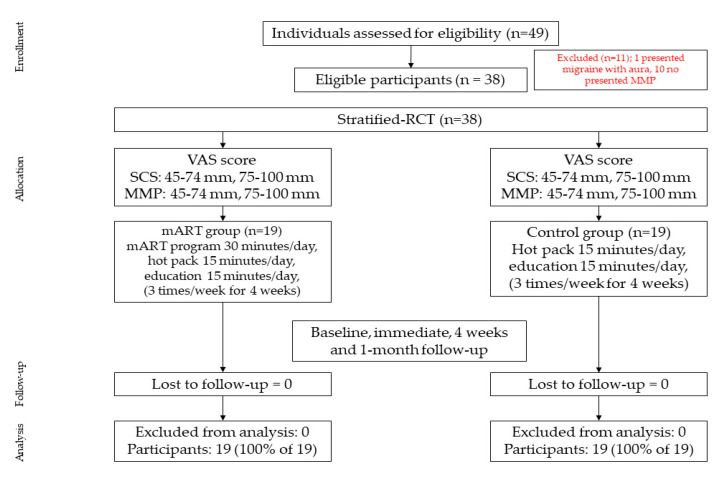
Flow diagram displaying stratified-randomized controlled trial participant pathways throughout the 4-week study.

**Table 1 ijerph-18-08533-t001:** Participant’s demographic data and health status.

Characteristics	mART Group (*n* = 19)	Control Group (*n* = 19)	*p*-Value
Age (years)	26 ± 5	25 ± 5	0.671 ^a^
Weight (kg)	58.74 ± 11.69	54.32 ± 10.58	0.679 ^a^
Height (cm)	164.16 ± 7.69	160.47 ± 6.62	0.435 ^a^
Body mass index (kg/m^2^)	21.72 ± 3.39	21.04 ± 3.50	0.984 ^a^
Gender			1.00 ^b^
Female	17 (89.5%)	17 (89.5%)	
Male	2 (10.5%)	2 (10.5%)	
Marital status			0.348 ^b^
Single	19 (100%)	17 (89.5%)	
Married	0	1 (5.3%)	
Widow/Divorced/Separated	0	1 (5.3%)	
Working age (months)	55.64 ± 56.86	55.80 ± 41.04	0.980 ^a^
Experienced pain duration (months)			
	30.63 ± 40.94	29.74 ± 27.51	0.634 ^a^
Location of pain (SCS) (*n*)			1.00 ^b^
Levator scapulae	16 (84.2%)	16 (84.2%)	
Rhomboids	3 (15.8%)	3 (15.8%)	
Serratus posterior superior	0	0	
Location of pain (MMP) (*n*)			1.00 ^b^
Masseter	19 (100%)	19 (100%)	
Temporalis	0	0	
Pterygoids	0	0	
Aggravating factors			0.400 ^b^
Typing	12 (63%)	12 (63%)	
Exercise	1 (5.3%)	2 (10.6%)	
Prolong sitting	3 (15.8%)	1 (5.3%)	
Lifting	1 (5.3%)	4 (21.1%)	
Compression force	1 (5.3%)	0	
Mouse use	1 (5.3%)	0	
Pain radiating (one person reported > 1 symptom)			0.458 ^b^
Non	8 (32.2%)	7 (26.9%)	
Headache	8 (25%)	3 (11.5%)	
Chest wall	5 (17.9%)	4 (15.4%)	
Eye	2 (7.1%)	7 (26.9%)	
Neck	2 (7.1%)	3 (11.5%)	
Back	2 (7.1%)	1 (3.9%)	
Jaw	1 (3.6%)	1 (3.9%)	

Notes: The data are presented using mean ± standard deviation and *n* (%), ^a^ The differences between the mART group and control group were compared using independent *t*-tests with the level of difference significance set at *p* < 0.05; ^b^ The variations between the mART group and control group were compared utilizing the chi-square test with the level of significant difference set at *p* < 0.05.

**Table 2 ijerph-18-08533-t002:** The comparison of outcomes within and between the 2 groups at baseline, immediate, 4 weeks, and 1-month follow-up.

Variables	Group	Baseline	Immediate	4 Weeks	1-Month Follow-Up
VAS_SCS_ ^#^	mART	4.98 ± 1.69	4.33 ± 1.68 *	2.52 ± 1.78	2.25 ± 1.74 *
	Control	5.71 ± 1.85	5.84 ± 2.18 *	4.63 ± 2.64	3.75 ± 2.34 *
VAS_MMP_ ^#^	mART	4.71 ± 1.70	4.18 ± 1.70	2.02 ± 1.47 *	2.90 ± 1.86 *
	Control	5.78 ± 2.27	5.63 ± 2.39	3.88 ± 2.76 *	3.23 ± 2.24 *
PPT_SCS_ ^#^	mART	1.56 ± 0.90	1.39 ± 0.75	2.28 ± 0.86 *	2.27 ± 0.88
	Control	1.45 ± 0.58	1.15 ± 0.45	1.72 ± 0.50 *	1.55 ± 0.78
PPT_MMP_ ^#^	mART	0.56 ± 0.37	0.56 ± 0.28	1.03 ± 0.26 *	0.78 ± 0.42
	Control	0.57 ± 0.31	0.55 ± 0.28	0.77 ± 0.26 *	0.58 ± 0.43
MO	mART	33.57 ± 4.55	35.61 ± 4.17	36.94 ± 4.08	35.13 ± 4.72
	Control	34.86 ± 5.19	33.63 ± 4.43	35.65 ± 3.93	36.46 ± 4.62
MMO	mART	41.10 ± 5.90	42.09 ± 6.41	42.81 ± 5.27	42.89 ± 4.63
	Control	41.40 ± 4.91	41.09 ± 4.15	42.72 ± 5.68	42.70 ± 5.33
CV-angle	mART	38.27 ± 5.11	39.06 ± 5.79	37.36 ± 3.72	37.91 ± 5.45
	Control	37.39 ± 5.10	36.51 ± 4.61	35.01 ± 5.15	36.56 ± 4.70
PCS	mART	22.16 ± 9.33	N/A	12.35 ± 10.11 *	10.46 ± 10.45 *
	Control	20.24 ± 11.41	N/A	13.16 ± 9.77 *	12.42 ± 10.49 *

Note: The data are presented using mean ± SD. Two-way mixed measure ANOVA * = Significant difference between immediate, 4 weeks, and 1-month follow-up with baseline set at *p* < 0.01; # = Significance between the two groups; Abbreviations: VAS = Visual analog scale, PPT = Pressure pain threshold, MO = Mouth opening, MMO = Maximum mouth opening, CV-angle = Craniovertebral angle, PCS = Pain catastrophizing scale, SCS = Scapulocostal syndrome, MMP = Masticatory myofascial pain syndrome, N/A = Not assessment.

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
