# Peer review of "Modified-Active Release Therapy in Patients with Scapulocostal Syndrome and Masticatory Myofascial Pain: A Stratified-Randomized Controlled Trial"

_ijerph, 2021, doi:10.3390/ijerph18168533_

Round 1
Reviewer 1 Report
- Recommendation:
Minor revision
- Comments to Author:
Manuscript ID: ijerph-1253991
Title: Modified-Active Release Therapy in Patients with Scapulocos-tal Syndrome and Masticatory Myofascial Pain: A Strati-fied-Randomized Controlled Trial
I found the paper to be overall well written and well described. I consider that the current study is relevant and of general interest to the readers of the journal.
- Minor comments:
Section 2.1. Study design and participants. I would add here the information about the participants (mainly age and sex). It is mentioned in the results in more detail but it should be also included in this section (eg. This study was a parallel design with stratified-randomized (block 4 and 6) control trial on thirty-eight academic participants (4 males and 34 females with ages ranging from….).
Is there any reason to have more females than males? If there is any this information should also be explained in this section.
Author Response
Dear Reviewers,
On behalf of all authors, I would like to sincerely thank reviewers for valuable time, comments and suggestions which improve the quality and clarity of the paper. In this version, all revisions have been made according to the suggestion as indicated using yellow highlight and explained in the rebuttal latter. Moreover, because of reviewers required grammar correct. thus, we sent the manuscript to the native English speaker to recheck all of grammar and shown as the track changes.
Response to Reviewer 1 Comments
Point 1: I found the paper to be overall well written and well described. I consider that the current study is relevant and of general interest to the reader of the journal.
Response 1: On behalf of all authors, I would like to sincerely thank you for all kind of support, valuable time, comments, and suggestions. All of the comments are helpful for improve the quality and clarity of the paper. Therefore, we are doing our best to answer and clarify all of the points followed the suggestion from the reviewers.
Point 2: Section 2.1 Study design and participants. I would add here the information about the participants (mainly age and sex). It is mentioned in the results in more detail, but it should be also included in this section (eg. This study was a parallel design with stratified-randomized (block 4 and 6) control trial on thirty-eight academic participants (4 males and 34 females with age ranging from....)).
Response 2: The information about the participants were added as shown in Line 82-83
Point 3: Is there any reason to have more females than males? If there is any this information should also be explained in this section.
Response 3: The additional information was added in text as shown in Line 83-86, and also added in the reference as shown in Line 961-964

Reviewer 2 Report
Dear Author,
I am honored to review your paper.
I think your research thesis is quite novel and interesting.
Although your manuscript covers a very interesting topic, there are some logical flaws when looking at the results of your paper.
In the inclusion criteria of this study, it was judged only with the palpation of the masticatory muscle and the amount of opening as the criteria for diagnosing MMP. However, although it was said that osteoarthritis patients were excluded from this study, but they were not excluded through actual radiographic examination. It is clear that muscle palpation is the most important factor in diagnosing muscular diseases among temporomandibular disorders, radiological examination is necessary to evaluate that it is not affected by other factors.
Your study showed that mART program was effective in controlling pain and improving emotional factors (pain catastrophizing) in SCS and MMP patients. However, it was not statistically significantly more effective than conventional home care therapy(hot packs and an educational briefing). Therefore, it is difficult to conclude that mART therapy must be considered in treating mcs and MMP patients.
I expect that you will conduct your research by complementing the study design to achieve logically valid results.
Thank you.
Author Response
Dear Reviewers,
On behalf of all authors, I would like to sincerely thank reviewers for valuable time, comments and suggestions which improve the quality and clarity of the paper. In this version, all revisions have been made according to the suggestion as indicated using yellow highlight and explained in the rebuttal latter. Moreover, because most of reviewers required grammar correct, thus, we sent the manuscript to the native English speaker to recheck all of grammar and shown as the track changes.
Response to Reviewer 2 Comments
Point 1: I am honored to review your paper. I think. your research is quite novel and interesting. Although your manuscript covers a very interesting topic, there are some logical flaws when looking at the results of your paper.
Response 1: On behalf of all authors, I would like to sincerely thank you for all kind of support, valuable time, comments, and suggestions. All of the comments are helpful for improve the quality and clarity of the paper. Therefore, we are doing our best to answer and clarify all of the points followed by the suggestion from the reviewers.
Point 2: In the inclusion criteria of this study, it was judged only with the palpation of the masticatory muscle and the amount of opening as the criteria for diagnosis MMP. However, although it was said that osteoarthritis patients were excluded from this study, but they were not excluded through actual radiographic examination. It is clear that muscle palpation is the most important factor in diagnosing muscular diseases among temporomandibular disorders, radiological examination is necessary to evaluate that it is not affected by other factors.
Response 2: Thank for your kind suggestion, it is interesting and important. Unfortunately, the data collection of this study already complete. Therefore, this suggestion was added in text for the further study.
Point 3: Your study showed that mART program was effective in controlling pain and improving emotional factors (pain catastrophizing) in SCS and MMP patients. However, it was not statistically significant more effective than conventional home care therapy (hot packs and an educational briefing). Therefore, it is difficult to conclude that mART therapy must considered in treating mcs and MMP patients.
Response 3: The results of the study revealed that pain intensity and pressure pain threshold were significantly improved in mART greater than the control group (as shown in Table 2 and Line 482, 489, 514, and 520). Therefore, it could be concluded that mART can be considered as an alternative treatment for relief pain in patients with SCS and MMP. However, the PCS did not show statistical significance between two groups, even though previous study mention about the association between pain intensity and pain catastrophizing. To clarify, pain catastrophizing is a perception of pain which depends on pain experience among individuals. According to pain experience and coping with pain among individuals are unique, the PCS questionnaires may not enough to answer the association of pain relief and pain perception among these participants. Therefore, the additional method or procedure should be conducted on pain perception for the future study research.

Reviewer 3 Report
The proposed article „ Modified-Active Release Therapy in Patients with Scapulocos- 2
tal Syndrome and Masticatory Myofascial Pain: A Stratified- 3
Randomized Controlled Trial “focuses on small problem of wide population. The techniques used in experiment are well known, but the combination is unique. The scapular and costal pain are frequently linked with another musculoskeletal problems and the tension of masticatory muscles are one of them. I would expect more tight relations in those two units. The modified therapy was successful in pain release, and that is an interesting result. The methods are sound, supported by adequate statistics. Finally, I guess this article might be interesting for the readers.
Author Response
Dear Reviewers,
On behalf of all authors, I would like to sincerely thank reviewers for valuable time, comments and suggestions which improve the quality and clarity of the paper. In this version, all revisions have been made according to the suggestion as indicated using yellow highlight and explained in the rebuttal latter. Moreover, because most of reviewers required grammar correct, thus, we sent the manuscript to the native English speaker to recheck all of grammar and shown as the track changes.
Response to Reviewer 3 Comments
Point 1: Randomized Controlled Trial "focuses on small problem of wide population". The techniques used in experiment are well know, but the combination is unique. The scapular and costal pain are frequently linked with another musculoskeletal problems and the tension of masticatory muscles are one of them. I would expect more tight relations in those two units. The modified therapy was successful in pain release, and that is an interesting result. The methods are sound, supported by adequate statistics. Finally, I guess this article might be interesting for the readers.
Response 1: On behalf of all authors, I would like to sincerely thank you for all kind of support, valuable time, comments, and suggestions. All of the comments are helpful for improve the quality and clarity of the paper. Therefore, we are doing our best to answer and clarify all of the points followed by the suggestion from the reviewers. In case of this question, the additional information for explaining the relation between SCS and MMP was added in the text as shown in Line 55-58

Reviewer 4 Report
This is a well-conducted study about the effectiveness of the mART in treating pain, promoting function, and measure emotions in patients with scapulocostal syndrome and masticatory myofascial pain. Material and methods are appropriate and results are deeply discussed and analysed. However, some changes are required in order to improve the work.
The introduction is well developed but written with so long periods and phrases (for example from line 55 to 61 there is only one period). Athors should reduce the lenght of so many phrases in order to make introduction clearer.
Line 36: the concept could be better expressed by reducing the presence of word “pain” into the phrase or by making two different phrases.
In the introduction, acronimum of scapulocostal syndrome is used but it was not indicated that it referred to that pathology. Please specify as soon as you used this acronimum.
Line 72-73: please make the concept clearer
In materials and methods, please clarify that the treatment of the control group represented the actual gold standard for that pathologies.
Line 304-305: neural 304 adaptation occurred before the muscular adaptation. Please put a reference
Author Response
Dear Reviewers,
On behalf of all authors, I would like to sincerely thank reviewers for valuable time, comments and suggestions which improve the quality and clarity of the paper. In this version, all revisions have been made according to the suggestion as indicated using yellow highlight and explained in the rebuttal latter. Moreover, because most reviewers required grammar correct, thus, we sent the manuscript to the native English speaker to recheck all grammar and shown the track changes.
Response to Reviewer 4 Comments
Point 1: This is a well-conducted study about the effectiveness of the mART in treating pain, promoting function, and measure emotions in patients with scapulocostal syndrome and masticatory myofascial pain. Material and methods are appropriate, and results are deeply discussed and analyzed. However, some changes are required in order to improve the work.
Response 1: On behalf of all authors, I would like to sincerely thank you for all kind of support, valuable time, comments, and suggestions. All of the comments are helpful for improve the quality and clarity of the paper. Therefore, we are doing our best to answer and clarify all of the points followed by the suggestion from the reviewers.
Point 2: The introduction is well developed but written with so long periods and phrases (for example from Line 55-61 there is only one period). Authors should reduce the length of so many phrases in order to make introduction clearer.
Response 2: This section was re-written as shown in Line 70-71
Point 3: Line 36: The concept could be better expressed by reducing the presence of word "pain" into the phrase or making two different phrases.
Response 3: This section was revised as shown in Line 35-37.
Point 4: In the introduction, acronym of scapulocostal syndrome is used but it was not indicated that it referred to that pathology. Please specify as soon as you used this acronym.
Response 4: The scapulocostal syndrome (SCS) was added in Line 60 as reference for using SCS as acronym.
Point 5: Line 72-73: please make the concept clearer
Response 5: The concept of using RCT was added as shown in Line 83-86
Point 6: In materials and methods, please clarify that treatment of the control group represented the actual gold standard for that pathologies.
Response 6: The additional information was added to clarify that the treatment program of control group represented the actual standard for these pathologies as shown in Line 153-157 and also added in the reference as shown in Line 970-971.
Point 7: Line 304-305: neural 304 adaptation occurred before the muscular adaptation. Please put a reference
Response 7: The additional reference was added as shown in Line 635 and also added in the reference as shown in Line 1030

Round 2
Reviewer 2 Report
After reviewing the manuscript, I could confirm that the items suggested for correction were well reflected.Therefore, this manuscript could be published in this journal. I am pleased to be offered the opportunity to review this paper. Thank you.